# Avian Influenza Virus and Avian Paramyxoviruses in Wild Waterfowl of the Western Coast of the Caspian Sea (2017–2020)

**DOI:** 10.3390/v16040598

**Published:** 2024-04-12

**Authors:** Tatyana Murashkina, Kirill Sharshov, Alimurad Gadzhiev, Guy Petherbridge, Anastasiya Derko, Ivan Sobolev, Nikita Dubovitskiy, Arina Loginova, Olga Kurskaya, Nikita Kasianov, Marsel Kabilov, Junki Mine, Yuko Uchida, Ryota Tsunekuni, Takehiko Saito, Alexander Alekseev, Alexander Shestopalov

**Affiliations:** 1Federal Research Center of Fundamental and Translational Medicine, Siberian Branch, Russian Academy of Sciences (FRC FTM SB RAS), Novosibirsk 630060, Russia; murashkinatatiana89@gmail.com (T.M.); a.derko19@gmail.com (A.D.); sobolev.riov@yandex.ru (I.S.); nikitadubovitskiy@gmail.com (N.D.); loginova995@gmail.com (A.L.); kurskaya_og@mail.ru (O.K.); naukanestoitnameste@mail.ru (N.K.); al-alexok@yandex.ru (A.A.); shestopalov2@mail.ru (A.S.); 2Faculty of Ecology and Sustainable Development, Dagestan State University, Makhachkala 367016, Russia; ali-eco@mail.ru; 3Caspian Centre for Nature Conservation, International Institute of Ecology and Sustainable Development, Association of Universities and Research Centres of Caspian Region States, Makhachkala 367016, Russia; caspianconservation@mail.ru; 4Institute of Chemical Biology and Fundamental Medicine, Novosibirsk 630090, Russia; kabilov@niboch.nsc.ru; 5Division of Transboundary Animal Disease, National Institute of Animal Health, Tsukuba 305-0856, Japan; minejun84032@affrc.go.jp (J.M.); uchiyu@affrc.go.jp (Y.U.); tune@affrc.go.jp (R.T.); taksaito@affrc.go.jp (T.S.)

**Keywords:** avian influenza viruses, phylogenetics, avian paramyxoviruses, avian avulaviruses, NDV, APMV-4, APMV-6, surveillance, wild waterfowl, migration, coastal area, Caspian Sea

## Abstract

The flyways of many different wild waterfowl pass through the Caspian Sea region. The western coast of the middle Caspian Sea is an area with many wetlands, where wintering grounds with large concentrations of birds are located. It is known that wild waterfowl are a natural reservoir of the influenza A virus. In the mid-2000s, in the north of this region, the mass deaths of swans, gulls, and pelicans from high pathogenicity avian influenza virus (HPAIV) were noted. At present, there is still little known about the presence of avian influenza virus (AIVs) and different avian paramyxoviruses (APMVs) in the region’s waterfowl bird populations. Here, we report the results of monitoring these viruses in the wild waterfowl of the western coast of the middle Caspian Sea from 2017 to 2020. Samples from 1438 individuals of 26 bird species of 7 orders were collected, from which 21 strains of AIV were isolated, amounting to a 1.46% isolation rate of the total number of samples analyzed (none of these birds exhibited external signs of disease). The following subtypes were determined and whole-genome nucleotide sequences of the isolated strains were obtained: H1N1 (*n* = 2), H3N8 (*n* = 8), H4N6 (*n* = 2), H7N3 (*n* = 2), H8N4 (*n* = 1), H10N5 (*n* = 1), and H12N5 (*n* = 1). No high pathogenicity influenza virus H5 subtype was detected. Phylogenetic analysis of AIV genomes did not reveal any specific pattern for viruses in the Caspian Sea region, showing that all segments belong to the Eurasian clades of classic avian-like influenza viruses. We also did not find the amino acid substitutions in the polymerase complex (PA, PB1, and PB2) that are critical for the increase in virulence or adaptation to mammals. In total, 23 hemagglutinating viruses not related to influenza A virus were also isolated, of which 15 belonged to avian paramyxoviruses. We were able to sequence 12 avian paramyxoviruses of three species, as follows: Newcastle disease virus (*n* = 4); Avian paramyxovirus 4 (*n* = 5); and Avian paramyxovirus 6 (*n* = 3). In the Russian Federation, the Newcastle disease virus of the VII.1.1 sub-genotype was first isolated from a wild bird (common pheasant) in the Caspian Sea region. The five avian paramyxovirus 4 isolates obtained belonged to the common clade in Genotype I, whereas phylogenetic analysis of three isolates of Avian paramyxovirus 6 showed that two isolates, isolated in 2017, belonged to Genotype I and that an isolate identified in 2020 belonged to Genotype II. The continued regular monitoring of AIVs and APMVs, the obtaining of data on the biological properties of isolated strains, and the accumulation of information on virus host species will allow for the adequate planning of epidemiological measures, suggest the most likely routes of spread of the virus, and assist in the prediction of the introduction of the viruses in the western coastal region of the middle Caspian Sea.

## 1. Introduction

The influenza virus is one of the best-known pathogens, having a significant impact on public health and the global economy. According to the latest data of the International Committee on Taxonomy of Viruses (ICTV), the influenza virus is represented by the following four genera: Alphainfluenzavirus, Betainfluenzavirus, Deltainfluenzavirus, and Gammainfluenzavirus [1]. *Alphainfluenzavirus* (influenza virus A) has the greatest epidemiological significance and can be the cause of seasonal outbreaks and sometimes pandemics [2]. The influenza A virus is generally characterized as ‘multi-host’, that is, it has an ability to infect various species of mammals, including humans, horses, pigs, seals, and birds, causing epizootics primarily among the latter. This causes serious economic damage to poultry farming, resulting in the mandatory destruction of large numbers of farmed birds [3,4].

The avian influenza virus (AIV) can overcome the interspecies barrier because of its high rate of genome variability through reassortment, which can lead to the emergence of strains that are highly pathogenic for humans, such as H5N1. This strain is currently of considerable concern, as it has already spread rapidly in five continents, affecting wild and domestic birds and mammal species. According to the World Health Organization, during the monitoring period from 2003 to 2023, 868 cases of human infection with the H5N1 avian influenza virus were documented, 457 of which resulted in death [5].

In wild birds, which are a natural reservoir of the influenza A virus, the disease, as a rule, is not accompanied by pronounced clinical signs. Being asymptomatic, this facilitates the transfer of the pathogen over significant distances, particularly during seasonal migrations. Of all the subtypes known to date, 16 hemagglutinin subtypes and 9 neuraminidase subtypes, in various combinations, have been isolated from wild birds. The influenza A virus has been found more often in representatives of the *Anseriformes* and *Charadriiformes* orders, than in others [6,7].

In 2009–2010, the existence of new subtypes of influenza A was recognized, which were isolated from frugivorous representatives of the order *Chiroptera* in Guatemala (influenza A virus subtypes H17N10 and H18N11), which have the potential to reassort with human influenza viruses [8]. Recently, genomic evidence was reported for a new candidate HA subtype, nominally H19, with a large genetic distance to all previously described AIV subtypes [9].

Until the recent spread of the H5N1 zoonosis amongst mammals, the scientific monitoring of avian influenza viruses has been carried out primarily in the natural habitats of those wild bird species of the highest epidemiological importance in the spread of these viruses. Of particular significance, in the context of this study, are the various wetlands and coastal waters of the western region of the middle Caspian Sea basin, which serves as a flyway corridor for birds migrating between the Mediterranean basin, Africa, and Eurasia and provides suitable loci for mass gatherings of birds; for nutritional stopovers; breeding; and wintering [10].

Previous studies conducted in the Caspian region have detected the circulation of various influenza A viruses, for example in water birds on the southern coast of the Caspian Sea in Iran [11], as well as in neighboring Georgia [12].

Many outbreaks of highly pathogenic subtypes of avian influenza (HPAIV) have been recorded in the northern Caspian region among wild avifauna; the first HPAIV outbreak, with mass deaths of at least 400 mute swans, occurred in 2005 [13] and the most recent occurred in 2022 of more than 30,000 gulls, terns, and pelicans [14]. In addition to influenza A viruses, there are other viruses associated with wild waterfowl. These are the avian paramyxoviruses (APMVs) belonging to subfamily Avulavirinae, order Paramyxoviridae [1], the first detection of which was in the Caspian region, where avian paramyxovirus APMV-4 was discovered in the western middle Caspian littoral [15].

All the above facts determined the relevance and purpose of this present study—to assess the genetic diversity, ecological features, and basic biological properties of avian influenza viruses and Avian paramyxoviruses in wild bird populations on the western coast of the middle Caspian Sea. The field and related laboratory work was undertaken in the period of 2017–2020, as described below.

## 2. Materials and Methods

### 2.1. Ethical Issues

This study was conducted with the approval of the Biomedical Ethics Committee of FRC FTM SB RAS, Novosibirsk (Protocols No. 2019-3 and 2021-10). The bird specimens were collected during the hunting season, with a license from the regional Ministries of Ecology and Natural Resources, as part of the annual collection of biological material (i.e., the Program for the Study of Infectious Diseases of Wild Animals, FRC FTM, Novosibirsk).

### 2.2. Sample Collection

Cloacal swabs of freshly hunted wild waterfowl were collected during the official hunting season in individual 2 mL tubes filled with 1 mL of viral transport medium without glycerol, consisting of phosphate-buffered saline (PBS, pH 7.5), amphotericin B (15 µg/mL), penicillin G (100 units/mL), and streptomycin (50 µg/mL). The tubes were immediately stored in liquid nitrogen and were transported to the FRC FTM SB RAS laboratory for analysis [16]. The collection of material was undertaken during the autumn–winter periods of 2017–2020, in places of the highest density of bird populations in the Republic of Dagestan, as follows: in the wetlands of Lake Adzhi (Papas), Lake South Agrakhan, Agrakhan Bay, the Terek River delta, and the Achikol system of lakes. Sampling points were selected based on previously compiled data from key ornithological territories [10].

### 2.3. Avian Influenza Virus and Avian Paramyxovirus Isolation Using Chicken Embryos

For the isolation of AIVs and APMVs, an aliquot was taken for each sample, was mixed using a vortex shaker, and then the suspension was centrifuged for 3 min at 3000× *g*. Antibiotics (penicillin and gentamicin) were added to the supernatant, which was transferred to a new test tube to avoid bacterial infection. Next, 100 µL of each sample were inoculated into the allantoic cavity of two specific pathogen-free (SPF) chicken embryos and incubated at 37 °C for 72 h in our Biosafety level-3 (BSL-3) laboratory [17]. After cultivation, 2 mL of each allantoic fluid were extracted and used for a hemagglutination test (HA) with 5% chicken red blood cells [18]. All HA-positive samples were aliquoted for AIV M gene and APMV RNA-dependent RNA polymerase (RdRp) gene PCR testing.

### 2.4. RNA Extraction, Reverse Transcription, and PCR

#### 2.4.1. Avian Influenza Virus Detection

All samples with HA activity were tested for the presence of influenza A. For this, RNA was extracted from the allantoic fluid using the RIBO-sorb kit (AmpliSens, Moscow, Russia). The resulting RNA was used in the reverse transcription reaction using the REVERTA-L kit (AmpliSens, Russia). The presence of conserved M gene regions of the influenza A virus was determined using real-time PCR using the AmpliSens Influenza virus A/B-FL kit (AmpliSens, Moscow, Russia), developed to detect both human and avian influenza virus.

#### 2.4.2. Avian Paramyxovirus Detection

HA-positive allantoic fluid was also tested for the presence of APMVs using RT-PCR. For this purpose, RNA was isolated and cDNA was synthesized, as described above. APMVs were detected using family wide oligonucleotides specific to the conserved region of the viral polymerase [19]. The reaction mixture contained 25 μL of Quick-Load Taq 2x Master Mix (New England Biolabs, Ipswich, MA, USA), 1 μL of forward and reverse oligonucleotides at a concentration of 50 pmol/μL each, 18 μL of water, and 5 μL of cDNA. The reaction mixture was incubated at 94 °C for 1 min, followed by 40 cycles at 94 °C for 15 s, 41 °C for 30 s, 68 °C for 30 s, and a final extension at 68 °C for 7 min.

Electrophoresis in 1.5% agarose gel was used to visualize PCR products. If a target fragment (121 nt) was detected, samples were prepared for whole-genome sequencing.

### 2.5. Intravenous Pathogenicity Index in Chickens

All animal experiments were conducted in Biosafety level-3 facilities and were approved by the Ethics Committee of the Federal Research Center of Fundamental and Translational Medicine (No. 2019-3).

For the intravenous pathogenicity index (IVPI) test, which was performed and calculated according to the WOAH standard protocol [17], we used 10 six-week-old specific pathogen-free white Leghorn chickens. Chickens were intravenously inoculated with 0.1 mL of 1:10 diluted infectious allantoic fluid (containing 10^6.0^ EID_50_ of the virus) and were monitored daily for clinical signs and were monitored for mortality for 10 days. The pathogenicity index, IVPI, was calculated according to the protocol as the mean score per bird per observation.

On day 14, post-inoculation (p.i.) blood samples were collected from chickens, which all survived, and sera were harvested and stored in laboratory collection for further possible studies and antigenic analysis.

### 2.6. Sequencing of AIV and APMV

For complete genome sequencing of the viruses collected between 2017 and 2018, RNA was isolated from allantoic fluids using the GeneJET viral DNA/RNA purification kit (Thermo Fisher Scientific, Waltham, MA, USA) and treated with TURBO DNase (Thermo Fisher Scientific, USA). Up to 200 ng of RNA was used for preparation of the DNA libraries, using the NEBNext Ultra RNA Library Prep kit (New England Biolabs, USA). Sequencing of the DNA libraries was conducted with a Reagent kit Version 3 (600-cycle) using the MiSeq genome sequencer (Illumina, San Diego, CA, USA) in the Genomics Core Facility (ICBFM SB RAS, Novosibirsk, Russia). Full-length genomes were assembled de novo with CLC Genomics Workbench v.9.5.3 (Qiagen, Hilden, Germany). The raw data of the sequences that were used for assembly have not been deposited and are stored and available in the “Genomics Core Facility” (ICBFM SB RAS, Novosibirsk, Russia).

Some of the strains were sequenced as part of a joint international program with the National Institute of Animal Health (Tsukuba, Japan). Thus, isolation of RNA from allantoic fluid was carried out using the RNeasy Mini kit (QIAGEN, Hilden, Germany). We used the NEBNext Ultra II RNA Library Prep Kit for Il-lumina (New England Biolabs, Ipswich, MA, USA) to prepare cDNA libraries. In total, 10 pM of libraries were mixed with 10 pM of PhiX Control V3 (Illumina) before sequencing. Sequencing was performed using the MiSeq genome sequencer (Illumina), using the MiSeq Reagent Kit v.2 (Illumina). Consensus sequences were constructed using Workbench software v.9.5.3 (QIAGEN, Germany). The most relevant and reference sequences from the U.S. National Center for Biotechnology Information (NCBI) GenBank database were included in the analysis.

### 2.7. Phylogenetic Analysis of Avian Influenza Virus

The subtyping of eight isolates was carried out using a BLAST analysis (Basic Local Alignment Search Tool, https://blast.ncbi.nlm.nih.gov/Blast.cgi last accessed on 15 March 2024) of the assembled HA and NA sequences. Specifically, the subtype was determined by the high percentage identity of the nucleotide sequences of the HA and NA gene of each isolate [20].

For each of the sequences of the segments of influenza viruses obtained, a search was carried out for identical or the most similar sequences using the BLAST algorithm (with an analysis depth of 50) from the GISAID database.

The resulting sequence sets were subjected to multiple alignment using the MAFFT v7.520 multiple alignment algorithm (downloadable version) [21] using the default parameters, as follows: gap opening penalty: 1.53 and gap extension penalty: 0.0. Files containing multiple alignments were then opened in the Unipro UGENE program [22], where sites containing deletions were manually checked and sites containing deletions were removed from the final alignment.

Using IQ-TREE software version 1.6 [23], phylogenetic trees were constructed using the ML (maximum likelihood) method and branch support was assessed using ultrafast bootstrap [24], as well as the SH-aLRT branch test with the number of iterations being equal to 1000. For each gene segment, the model of nucleotide substitutions was determined using ModelFinder through IQ-TREE version 1.6 [25].

Tree visualization and topology analysis were performed in iTOL v.6 [26].

In each of the six resulting phylogenetic trees, strains from different clades distributed throughout the tree were selected for better visualization. For example, phylogenetically similar repeats collected during the study in the same region and at the same time were excluded.

After this, the above operations, starting with multiple sequence alignment, were repeated again for the final sample.

Manipulations with files containing sequences were carried out, including with the aid of Biopython 1.81 [27] and the SeqKit toolkit [28].

### 2.8. Phylogenetic Analysis of Newcastle Disease Virus

For phylogenetic analysis, the complete F gene coding sequence (CDS) was used (1662 nt). To determine the genotype and sub-genotype, we used the data set described by Dimitrov et al. [29]. Multiple sequence alignments were produced using the MUSCLE alignment algorithm. The tree was constructed using the maximum likelihood method, based on the Tamura 3-parameter model with a discrete gamma distribution (+G); statistical analysis was based on 1000 bootstrap replicates, as implemented in MEGA X [30]. An additional tree was then constructed for the identified genotypes using the most closely related sequences found in BLAST. Multiple alignment and parameters for tree construction were the same as described above.

### 2.9. Phylogenetic Analysis of Avian Paramyxovirus 4

For phylogenetic analysis, the complete F gene CDS (1701 nt) was used. Multiple sequence alignments were produced using the MUSCLE alignment algorithm. The tree was constructed using the maximum likelihood method, based on the Tamura 3-parameter model with invariant sites (+I), with statistical analysis based on 1000 bootstrap replicates, as implemented in MEGA X [30].

### 2.10. Phylogenetic Analysis of Avian Paramyxovirus 6

For phylogenetic analysis, the complete F gene CDS (1668 and 1638 nt) was used. Multiple sequence alignments were produced using the MUSCLE alignment algorithm. The tree was constructed using the maximum likelihood method, based on the Tamura 3-parameter model with a discrete gamma distribution (+G) and allowing for invariant sites (+I); statistical analysis was based on 1000 bootstrap replicates, as implemented in MEGA X [30].

## 3. Results

### 3.1. Ecological Features of Study Area and Samples

As noted above, during 2017–2020, biological material from 1438 individual wild birds of 26 species on the western coast of the middle Caspian Sea was collected and studied. The collection activity took place over three periods. The first was from September 2017 to March 2018, the second was from September 2018 to March 2019, and the third was from September 2019 to March 2020.

The choice of these annual field collection seasons was determined on the basis of one of the principal ecological aspects of the faunal diversity of coastal waterfowl habitats, as determined via analysis of the literature and data from earlier reconnaissance expeditions. This aspect is that during this period of the year, there are significant, regular concentrations of wintering aquatic and semi-aquatic birds on the lagoons and coastal waters of the western coast of the middle Caspian Sea, which are zones providing temperate avian wintering conditions along migratory flyways [31]. The resultant contacts and sharing of habitats by large numbers of wild birds contribute to the spread of infectious diseases such as avian influenza viruses. In accordance with the above factors, we selected the collection points of the material (Figure 1).

The samples collected through this project belonged to wild birds of the following seven orders: *Anseriformes*, *Charadriiformes*, *Galliformes*, *Ciconiiformes*, *Gruiformes*, *Podicipediformes*, and *Pelicaniformes*. However, it should be noted that the sampled birds were mainly of two orders, *Anseriformes* and *Gruiformes*. The percentage of individual birds of these orders was 55.39% and 40.93% of the total number sampled, respectively. The remaining five orders of birds accounted for 3.68% of the total number sampled.

Of the 797 individuals of the order *Anseriformes* studied, there were mainly four species, as follows: mallard (*Anas platyrhynchos n* = 196), common teal (*Anas crecca n* = 158), garganey (*Anas querquedula n* = 102), and common pochard (*Aythya ferina n* = 138). The order *Gruiformes* was represented by two species, coot (*Fulica atra*) and common moorhen (*Gallinula chloropus*), with coot in the majority (*n* = 586).

All species sampled have a wide geographical distribution along the western coast of the middle Caspian Sea and migrate to the region to breed and, as the literature indicates, can also be vectors for the influenza A virus.

In addition to the principal species sampled, as given above, a small number of samples were also examined, which were collected from birds of less regionally represented species, such as shoveler (*A. clypeata*), northern pintail (*A. acuta*), European wigeon (*A. penelope*), greylag goose (*Anser anser*), and common shelduck (*Tadorna tadorna*). These *Anseriformes* species are also potential reservoirs of influenza viruses. Thus, their study and the detection of viruses carried by them is an important aspect of determining the natural reservoir of avian influenza pathogens.

The entire species diversity of wild birds, from which biological material was collected and studied in 2017–2020 on the western littoral of the middle Caspian Sea, is presented in Table 1.

### 3.2. Avian Influenza and Paramyxoviruses in Coastal Wild Birds of the Caspian Sea

During the 2017–2020 virological monitoring, 43 isolates of hemagglutinating agents were obtained from 1438 samples, which amounted to 2.99% of the total number of samples studied, 21 of which were identified as influenza A virus, thus amounting to 1.46% of the total number of samples studied. In total, 19 AIV isolates were obtained from individuals of the *Anseriformes* order (*Anatidae* family), constituting the largest share (83.7%) of all AIV isolates obtained. All representatives of the *Anatidae* sampled provided isolates. Most were obtained from mallard, garganey, and common teal, while six isolates were obtained from the coot (*Fulica atra*) of the rail family (*Rallidae*) and one isolate was obtained from the common pheasant (*Phasianus colchicus*) of the family *Phasianidae* (Figure 2).

A total of 15 hemagglutinating isolates related to avian paramyxoviruses (APMVs) were isolated, amounting to 1.04% of the total number of samples studied.

Of all the isolates obtained, 49% were avian influenza viruses, 35% were avian paramyxoviruses, but 16% of hemagglutinating agents could not be identified. Information about the isolates obtained is presented in Appendix A, which can be found in the Appendix A.

The results of subtyping determined that 21 isolated strains belonged to the following subtypes: HxNy (4), H1N1 (2), H3N8 (8), H4N6 (2), H7N3 (2), H8N4 (1), H10N5 (1), and H12N5 (1). Of all the strains identified, 38% were H3N8.

The birds from which the largest number of AIV isolates were obtained were common teal (*n* = 8). In total, six AIV isolates were obtained from mallard, eight AIV isolates from common teal, two AIV isolates from coot, and one isolate each was isolated from red-crested pochard and northern pintail.

Thus, according to the monitoring results for the period 2017–2020 (September to March collections), the isolation of influenza A virus amounted to 1.46% of the total number of samples studied.

The birds from which the largest number of isolates of hemagglutinating viruses were obtained in the first season belonged to common teal, amounting to nine individuals. Four isolates were isolated from mallard. Also, one isolate each was obtained from the following species: red-crested pochard, common teal, common pheasant, and northern pintail.

In the second season, the birds from which the largest number of isolates of hemagglutinating viruses were obtained belonged to common teal, amounting to five individuals, while three isolates were isolated from mallard, and two isolates from common teal. Also, one isolate each was obtained from coot, red-crested pochard, and northern pintail.

In the third season, the birds from which the largest number of isolates of hemagglutinating agents were obtained were coot, amounting to five individuals, while three isolates were isolated from mallard, two isolates each from common teal and garganey, and one isolate from tufted duck.

### 3.3. Phylogenetic Analysis of Avian Paramyxoviruses

In this study, we were able to sequence 12 avian paramyxoviruses of the following three species: Newcastle disease virus (*n* = 4), now referred to as the Orthoavulavirus; Avian paramyxovirus 4 (*n* = 5), now referred to as the Paraavulavirus [1]; and Avian paramyxovirus 6 (*n* = 3), now referred to as the Metaavulavirus [1] (Table 2).

#### 3.3.1. Newcastle Disease Virus Analysis

During the current study, four Newcastle disease virus isolates were identified. All of them belonged to Class II (Table 2). Three isolates belonged to sub-genotype I.2 and one belonged to sub-genotype VII.1.1 (Figure 3a), according to the current classification [29] used in this study.

Sub-genotype I.2

As a result of phylogenetic analysis of complete F gene CDS (1662 nt), it was found that the isolate NDV/common teal/Dagestan/Russia/54/2017 (PP537563) is most closely related to isolates from Western Siberia (Russia), Yakutia (Russia), Mongolia, and China. The isolate NDV/common teal/Dagestan/Russia/111/2017 (MZ666236) is related to isolates from Ukraine, Western Siberia, and Nigeria. The isolate NDV/mallard/Dagestan/Russia/28d/2019 (MW927498) was also phylogenetically close to isolates from Ukraine, Western Siberia, and China.

Amino acid analysis of protein F showed that all isolates contained the cleavage site of protein F 112-GKQGR↓L-117, which is a characteristic of the lentogenic variants of the virus.

Sub-genotype VII.1.1

The isolate NDV/common pheasant/Dagestan/Russia/33/2018 (PP537562) belonged to sub-genotype VII.1.1, according to [29]. This is the first case of detection of this sub-genotype in wild birds in Russia. (Figure 3b). Phylogenetic analysis has shown that this isolate is most closely related to highly pathogenic isolates and belongs to the cluster VII.1.1-l, formed by isolates from Iran.

As a result of the amino acid analysis of protein F, the cleavage site 112-RRQKR↓F-117 was established, which is characteristic of velogenic NDV variants.

#### 3.3.2. Avian Paramyxovirus 4 and 6

The five avian paramyxovirus 4 isolates obtained in the current study belonged to the common clade in Genotype I (Figure 4a). According to the results of phylogenetic analysis, they were most closely related to the viral variants found in Russia, in Western Siberia, Primorye and Buryatia, and in China. The isolate APMV-4/mallard/Dagestan/59d/Russia/2019 (MZ852793) also had a close phylogenetic relationship with an isolate isolated in Belgium in 2021.

For most isolates, the fusion protein cleavage motif was 116-DIQPR↓F-121. Only the isolate APMV-4/mallard/Dagestan/92d/2018 had the substitution of isoleucine with valine at position 117 (Table 2).

Three isolates of Avian paramyxovirus 6 were identified (Table 2). Phylogenetic analysis showed that two isolates isolated in 2017 belong to Genotype I and the isolate identified in 2020 belongs to Genotype II (Figure 4b).

The isolates APMV-6/common teal/Dagestan/Russia/62/2017 (PP537560) and APMV-6/common teal/Dagestan/Russia/130/2017 (PP537561) were the closest to isolates from Kazakhstan, Ukraine, China, and Korea. The F protein cleavage site was typical for apathogenic variants of the 114-VPEPR-L-119 virus (Table 2).

The isolate APMV-6/mallard/Dagestan/194d/Russia/2020 (PP537564) appeared in a clade with phylogenetically similar isolates from the Russian Far East region, Canada, the USA, and the eastern part of China. The protein cleavage site F 104-IREPR↓L-109 was also characteristic of apathogenic variants of the virus (Table 2).

### 3.4. Phylogenetic Analysis of Avian Influenza Virus

To construct phylogenetic dendrograms, we used sequences of genome segments of the influenza virus strains isolated, as well as closely related sequences from around the world, which were identified using BLAST analysis.

#### 3.4.1. Phylogenetic Analysis of Internal Segments of Genome

Phylogenetic analysis of internal segments showed that all are of Eurasian lineage. It is noteworthy that the NP segments of the strains studied belong to different clades, as follows: One clade contains NPs of various subtypes isolated in Europe and West Asia, as well as NPs of the highly pathogenic strain of the H5N8 subtype isolated in Korea. Another contains NPs of avian influenza viruses isolated, with the exception of the samples from Georgia, north Asia, east Asia, and south-east Asia. Additionally, both alleles of segment NS (A and B alleles) were detected among studied strains. A detailed description of the phylogenetic relationships of internal segments is presented in the Appendix A (Appendix A). We also assessed the major amino acid substitutions in polymerase complexes (PA, PB1, and PB2) that were previously described in the literature as significant for the mammalian adaptation of AIVs. We found that the PA of all strains in the study had S149 amino acid mutations associated with limited lethality in mice [32]. PB1 had L598 amino acid mutation that is responsible for replication decreasing in MDCK [33]. The protein PB2 contained 391E and 627E amino acid mutations in key positions, suggesting a decrease in the virulence in ferrets and a decrease in the replication in mammalian cells, respectively [34,35].

We have shown the presence of substitutions that do not increase the risk of adaptation to mammals. However, a limitation of such findings is that the role of such substitutions has been described for H5N1 influenza viruses, but has not been proved for LPAIs.

#### 3.4.2. Phylogenetic Analysis of Surface Glycoprotein Segments HA and NA

As the surface glycoproteins HA and NA are the main determinants of pathogenicity and antigenically relevant for vaccines, here we first analyzed the HA and NA segments of H1N1, H3N8, H4N6, H7N3, H8N4, H10N5, and H12N5, revealed in the Caspian Sea region.

Viruses A/mallard/Dagestan/1092/2018 and A/mallard/Dagestan/1093/2018 represent the subtype H1N1. According to the hemagglutinin H1 segment (Figure 5a), they belong to the Eurasian genetic lineage of avian influenza viruses and form one clade with AIVs circulated in the European part of the continent (Belgium, The Netherlands, and western Russia). A separate clade of the closest viruses from BLAST is formed by sequences of the HA segment of influenza viruses isolated in the Asian part.

The N1 neuraminidase gene sequence belonged to the Eurasian genetic lineage of avian influenza viruses and form a clade with “European” viruses, with the exception of one from Mongolia.

Viruses A/mallard/Dagestan/1050/2018 and A/mallard/Dagestan/1051/2018 representing the H7N3 subtype also belong to the Eurasian genetic lineage of avian influenza viruses according to the hemagglutinin H7 segment (Figure 6a). Interestingly, they form a single clade with viruses circulating in the Asian part of Eurasia. A separate clade is formed by sequences of the HA segment of influenza viruses isolated in North Africa (Egypt) and Georgia. For both H7N3 viruses, we showed the absence of basic amino acids at the HA cleavage site, indicating these viruses to be low pathogenic (PELPKGR/GLF). According to the neuraminidase N3 segment (Figure 6b), they also belong to the Eurasian genetic lineage of avian influenza viruses. Detailed characteristics of these Caspian H7N3 viruses have been described by us, in an earlier publication [36].

The sequence of the HA segment of the A/teal/Dagestan/57d/2018 (H8N4) virus (Figure 7a) belongs to the Asian clade and falls to the same clade as viruses from Bangladesh, Japan, and China, suggesting origination from or a relationship to Asian regions.

According to the N4 neuraminidase segment (Figure 7b), it also belongs to the Eurasian genetic lineage of avian influenza viruses. However, in this case, the sequence of the NA segment forms one clade with the sequence from Europe (Belgium), although this clade does not have reliable support.

The sequences of segments HA (Figure 8a) and NA (Figure 8b) of two H4N6 viruses, A/Common_Teal/Dagestan/34d/2019 and A/gadwall/Dagestan/156/2017, belong to the Eurasian genetic lineage and, according to the HA tree, form one clade with viruses from Bangladesh, Germany, and the Novosibirsk region of Russia. It suggests a wider connection with Eurasia. However, this clade does not have reliable branch support.

According to the neuraminidase N6 segment, they also belong to the Eurasian genetic lineage of avian influenza viruses. However, in this case, the studied samples are located in different clades. A/gadwall/Dagestan/156/2017 is closely related to the viruses from Belgium and Western Siberia (Novosibirsk region), but virus A/Common_Teal/Dagestan/34d/2019 is closely related to viruses from the Central Asia region (collected in Bangladesh).

All sequences of segments HA (Figure 9a) and NA (Figure 9b) of the eight H3N8 viruses studied belong to the Eurasian genetic lineage, which we divided formally into two clades. In general, the dendrogram contains H3N8 viruses mainly circulating in different Asian countries. A similar picture was found when analyzing the dendrogram of NA (N8).

All the studied segments HA (Figure 10a,b) and NA (Figure 10c) also belong to the Eurasian genetic lineage of Avian influenza viruses. The sequence of the HA segment of the A/mallard/Dagestan/004/2018, which has the H10 subtype, falls to the clade with viruses collected in Siberia (Novosibirsk Region). The A/teal/Dagestan/1017/2018 virus, which is of the hemagglutinin H12 subtype, is also related to the viruses collected at Lake Chany in Siberia; however, this clade also contains strains from Belgium and Poland.

Two neuraminidases N5 are not closely phylogenetically related to each other. However, it is interesting to note that the NA segment of the A/teal/Dagestan/1017/2018 (H12N5) virus shares one clade with Siberian H12N5 viruses, isolated from samples collected in the Lake Chany and Lake Ubinskoe systems—a very important breeding and stopover region for waterfowl in Asia.

In general, phylogenetic analysis of the surface protein segments did not reveal any specific pattern for viruses of the Caspian Sea region. All segments belong to the Eurasian clades of classic avian-like influenza viruses. This is consistent with the analysis of internal segments. The close phylogenetic relationships of some Caspian viruses with Asian and Siberian viruses on the one hand, and the close phylogenetic relationships with European viruses of other Caspian viruses on the other, suggest broad connections and ways of virus transmission throughout Eurasia. This is consistent with the data on the existence of numerous different flyways through the Caspian Sea basin of birds, which are hosts for such avian influenza viruses.

### 3.5. Intravenous Pathogenicity Index

Experimental infection of six-week-old specific pathogen-free chickens showed the absence of clinical signs of disease and deaths, including from the strains A/mallard/Dagestan/1092/2018 (H1N1), A/mallard/Dagestan/1093/2018 (H1N1), A/pochard/Dagestan/92/2017 (H3N8), A/teal/Dagestan/141/2017 (H3N8), and A/teal/Dagestan/44d/2018 (H3N8), in which, as a result of phylogenetic analysis, the NPs segments were found to be similar to those of the highly pathogenic subtype H5N8 from Korea.

## 4. Discussion

This summation of the results of the long-term monitoring of the influenza A virus in wild bird populations along the western littoral of the middle Caspian Sea is the first such undertaking. The high significance of the study area for the circulation of AIVs and APMVs among birds is determined by its location in an area of concentration of migratory flyways passing through a geomorphological “bottleneck” between the Black and Caspian seas and along the length of the Caspian basin itself [37].

Piedmont Dagestan, which occupies the north-eastern part of the northern macro-slope of the Greater Caucasus, has historically attracted the attention of ornithologists as an ecosystem which is biotopically differentiated and with varying precipitation characteristics, resulting in many variations in bird communities. Birds of the transitional ornithocomplexes of the foothills are interconnected with those of intramountain and high mountain Dagestan, as well as with the bird communities of the plains of the republic. Over a hundred species of Palearctic migratory birds which use the West Siberian–East African (or Eurasia–Africa) flyway pass through the main defiles of the Caucasus foothills, as the eastern flanks of the Caucasus mountains delineate a lowland zone along the western Caspian littoral, providing a substantial migration corridor for birds flying from further north or south, according to the season [38].

The nesting areas of these trans-Palaearctic migrant aquatic or semi-aquatic birds are concentrated in the Arctic, Subarctic, the West Siberian Plain, and Kazakhstan. Their primary wintering areas are located in the southern Caspian region (though—in response to climate change—some species are now wintering in the southern wetlands of the Dagestan coast), India (mainly for gulls), the Middle East, the Nile delta, and northeast Africa [39].

As noted above, along the Caspian part of the Republic of Dagestan, there are a number of aquatic habitats with high regional concentrations of aquatic or semi-aquatic populations of wild birds during seasonal migrations. Here, the intersection of migratory routes increases the likelihood of wild birds participating in the exchange of AIVs or APMVs over large geographical areas. The likelihood of the formation of virus variants with new antigenic properties within different populations and the risk of introducing strains of high pathogenicity avian influenza also increases through these eco-dynamics.

Dagestan habitats utilized by aquatic or semi-aquatic birds include shallow bays, estuaries of large rivers, lagoons, the coast itself, and off-shore islands. The wetlands of Lake South Agrakhan, Agrakhan Bay, Terek River delta, the Achikol lake system, and Lake Adzhi (Papas) are regularly used for authorized seasonal hunting of waterfowl. Lake Adzhi (Papas) is located approximately 250 km to the south of the other areas. The spatial separation of these various areas makes it possible to take samples from birds in different physical and geographical ecosystems of Caspian Dagestan. This is important for monitoring activities, as it ensures regional representation of target samples of species of birds monitored. In addition to a high diversity of species composition being a priority in choosing sampling sites, transport accessibility to these areas is also important, as this facilitates the efficient organization of sampling trips [10]. From the above, it is clear that the western coastal region of the middle Caspian Sea must play an important role in studies of the ecology of the influenza A virus in wild bird populations of aquatic ecosystems in Eurasia.

We have noted that during our long-term surveillance study among 1438 individuals of 26 bird species, we detected 21 AIV strains, amounting to a 1.46% isolation rate, most being H3N8 (43% of all AIVs). Here, it should be emphasized that in the present study, which monitored clinically healthy wild birds, no highly pathogenic H5Nx avian influenza was detected, although outbreaks have been reported in the region.

As our samples included not only ducks (which constitute the principal reservoir of such viruses), but also representatives of a number of other waterbird families, the percentage of AIV strains we detected was lower than indicated in similar studies of aquatic avifauna in neighboring biogeographical regions, in which avian influenza viral counts reflected a greater proportion of *Anatidae* sampled. For example, in the neighboring southern Caspian coast of Iran, AIVs were detected in approximately 3.4% of the samples. However, prevalence was higher (up to 8.3%) in some locations [11]. As in our present study, positive samples originated mainly from mallard and common teal, which are known principal vectors of AIVs.

In the neighboring region of Georgia, an increased AIV prevalence was also observed in ducks during the autumn postmolt aggregations and migration staging period. In this case, the AIV prevalence rate was found to be 6.3%, which is a higher rate than we found [12].

We were able to subtype and obtain complete genomes for 17 influenza viruses. Based on our phylogenetic analysis, we can conclude that the Caspian Sea littoral represents a unique region for the exchange of the genetic material of influenza viruses from various sources, including distant and disparate geographical regions. It is noteworthy that the strains isolated in Dagestan are genetically diverse and significantly different from each other. Individual segments of their genomes belong to different phylogenetic clades. In particular, according to the NS segment, the strains are clearly divided into two genetically distant phylogenetic clades.

Various strains from Dagestan are phylogenetically related in certain genome segments to influenza virus variants found in Siberia (Lake Chany and Lake Ubinskoe systems). In addition, genome segments similar to the genes of strains isolated in Europe (Germany, Czech Republic, The Netherlands, and Croatia) were identified.

Some of the genome segments analyzed are related to segments characteristic of East and Central Asia. Thus, among the strains studied, genetic material characteristics of various regions of Eurasia have been found.

In addition, similar segments of the influenza virus genome were found in Bangladesh (i.e., in south Asia), as well as in Egypt, the Sinai Peninsula, and northeast Africa. Genetically similar genome segments have also been identified in influenza viruses isolated in Georgia, which suggests the possibility of the spread of avian influenza viruses between the Caspian and Black Seas [40].

Differences in the phylogenetic environment of different segments of the same influenza virus strains isolated in Dagestan indicate reassortment events, leading to the emergence of virus variants with new combinations of genome segments.

It is noteworthy that some of the strains from Dagestan are related, in some genome segments, to the highly pathogenic variant of the influenza virus subtype H5N8. In particular, the strains A/pochard/Dagestan/92/2017 (H3N8) and A/teal/Dagestan/141/2017 (H3N8) are similar in the NP and PB1 segments to the strains of the H5N8 subtype and from different geographic regions (NP—from Asia, and PB1—from Europe), while the strain A/gadwall/Dagestan/156/2017 (H4N6) in the PB2 segment is related to strains of the H5N8 subtype from Bangladesh and Egypt. Thus, our study indicates the circulation of influenza virus variants that have arisen as a result of the exchange of genetic material between strains of different subtypes, including those which are highly pathogenic.

In summarizing the results of the phylogenetic analysis, we have determined that the avian influenza virus strains isolated in the Caspian Sea are genetically different from each other and form clusters of phylogenetically similar strains with influenza virus variants isolated throughout Eurasia, as well as in northeast Africa. Similar strains were isolated not only in the longitudinal direction or axis, which corresponds to migratory flight routes from north to south and vice versa, but also in the latitudinal direction. In the latter regard, the area of fixation of similar genomic segments in the latitudinal direction extends beyond individual flyways, which indicates mixing in large water areas and lake systems (as previously shown for the Lake Chany system, Western Siberia) of various genetic variants of the avian influenza virus, spreading along various flyways. However, for individual strains, the conservation of the distribution of genomic segments was also shown. For example, the nucleotide sequences of the genetic subclade of allele A of the phylogenetic dendrogram for the NS segment are characteristic of strains circulating in the east Asian region, while the NS sequences of the subclade of the same allele A are characteristic not only for Asian variants of the virus, but also for European ones. In addition, within the phylogenetic clades, the conservation of genome segments was shown throughout the entire period of work on the project. In the NP segment, in particular, strains A/teal/Dagestan/120/2017 and A/teal/Dagestan/57d/2018, as well as A/gadwall/Dagestan/156/2017 and A/teal/Dagestan/1017/2018, are pairwise maximally similar to each other, despite the fact that they were isolated from different seasons of sample collection.

As a result of our research, 15 avian paramyxoviruses (Paramyxoviridae, Avulavirinae) were identified in the Republic of Dagestan during the period 2017–2020. Of these, the species of virus was identified for 12 avian paramyxoviruses and genome-wide sequences were obtained.

The monitoring of avian paramyxoviruses in Russia is rare. However, recent results of a large-scale NDV study conducted from 2017 to 2021 [41] showed the presence of various NDV sub-genotypes in 28 regions of the country. The percentage of NDV release was 0.2% (15 out of 7410) for wild and synanthropic birds and 0.52% (63 out of 12,090) for domestic birds. In our study, this rate was 0.28% (4 out of 1438) (only wild birds were present in the samples). Guseva et al.’s study [41] identified strains of sub-genotype VII.1.1 in domestic birds. In our study, this sub-genotype was the first isolated from a wild bird (common pheasant) in Russia, in the Caspian Sea region.

Sub-genotype VII, Class II is represented by pathogenic variants of NDV detected in sick or dead wild and domestic birds. Strains of this genotype are currently predominant in the genotypes circulating in the world. Our NDV/common pheasant/Dagestan/Russia/33/2018 isolate belongs to sub-genotype VII.1.1, specifically to VII.1.1-l. Sub-genotype VII.1.1-l was formed by Iranian strains and probably appeared on the basis of strains of sub-genotype VII.1.1-d [42]. Strains of sub-genotype VII.1.1 were the cause of bird deaths in the Middle East, Asia, and Africa.

The sub-genotype VII.1.1 is responsible for ongoing outbreaks of Newcastle disease in Russia. Thus, in 2022, a mass death of birds was recorded at a backyard farm in the Moscow region [43]. The cause was a highly pathogenic strain of sub-genotype VII.1.1. Our results support the hypothesis that this sub-genotype can be spread by wild birds. The situation with the spread of the highly pathogenic NDV variant currently circulating in Russia may be complicated by the lack of vaccination of birds raised in domestic backyards. If wild birds such as the common pheasant do carry highly pathogenic variants of NDV, then unvaccinated poultry may become the foci of the spread of Newcastle disease to other regions of the Russian Federation.

Avian paramyxovirus 4 and Avian paramyxovirus 6 were also detected in our study. The isolation rate was 0.35% (5 out of 1438) and 0.21% (3 out of 1438), respectively. APMV-4 and APMV-6 are often detected during AIV monitoring in wild waterfowl in various regions of the world, including some neighboring the Caspian Sea [44,45,46]. It is currently unknown what role they play in the ecology of other viruses (for example AIVs). APMV-4 and APMV-6, which are probably non-pathogenic to their hosts, can interact with their immune system, making them immune to pathogenic variants of other APMVs.

Nevertheless, our long-term surveillance study has identified certain limitations that should be acknowledged. The main limitation is that we were unable to subtype several virus isolates and obtain their genome-wide sequences due to a lack of sequencing data quality. However, the complete genomes of 12 APMVs and 17 AIVs make a significant contribution to international databases for a further detailed analysis of viruses in the region, and Eurasia as a whole.

## 5. Conclusions

In conclusion, in 2017–2020, influenza A virus was monitored in wild bird populations of the western coast of the middle Caspian Sea. Samples from 1438 individuals of 26 bird species of 7 orders were collected, from which 21 strains of AIV were isolated, amounting to a 1.46% isolation proportion of the total number of samples analyzed. Strains were identified in representatives of the order *Anseriformes*, as follows: common teal (*Anas crecca n* = 8), mallard (*Anas platyrhynchos n* = 6), garganey (*Anas querquedula n* = 3), coot (*Fulica atra n* = 2), common pochard (*Aythya ferina n* = 1), and gadwall (*Anas strepera n* = 1). The following subtypes were determined and nucleotide sequences of isolated strains were obtained: H1N1 (*n* = 2), H3N8 (*n* = 8), H4N6 (*n* = 2), H7N3 (*n* = 2), H8N4 (*n* = 1), H10N5 (*n* = 1), and H12N5 (*n* = 1). Four isolates remained un-subtyped (HxNy, *n* = 4). Of all the identified strains, 43% were H3N8. No high pathogenicity influenza virus H5 subtype was detected. In total, 23 hemagglutinating viruses not related to influenza A virus were also isolated, of which 15 belonged to avian paramyxoviruses (APMVs). We were able to sequence 12 avian paramyxoviruses of the following three species: Newcastle disease virus (*n* = 4); Avian paramyxovirus 4 (*n* = 5); and Avian paramyxovirus 6 (*n* = 3). Newcastle disease virus of the VII.1.1 sub-genotype was first isolated from a wild bird (common pheasant) in Russia, of the Russian Federation in the Caspian Sea region. The five avian paramyxovirus 4 isolates obtained belonged to the common clade in Genotype I, whereas phylogenetic analysis of three isolates of Avian paramyxovirus 6 showed that two isolates isolated in 2017 belong to Genotype I and the isolate identified in 2020 belongs to Genotype II.

Phylogenetic analysis showed that all of these belong to Eurasian lineage. In general, phylogenetic analysis of internal segments and HA/NA segments did not reveal any specific pattern for viruses of the Caspian Sea region. All segments belong to the Eurasian clades of classic avian-like influenza viruses. The close phylogenetic relationships of some Caspian viruses with Asian and Siberian viruses on the one hand, and the close phylogenetic relationships with European viruses of other Caspian viruses on the other, suggest broad connections and ways of virus transmission throughout Eurasia. This is consistent with the data on the existence of numerous different flyways of birds passing through the Caspian Sea basin, which are the hosts of such avian influenza viruses. We also did not find the amino acid substitutions in polymerase complexes (PA, PB1, and PB2) that are critical for the increase in virulence or adaptation to mammals.

The continued regular monitoring of AIVs and APMVs, the obtaining of data on the biological properties of isolated strains, and the accumulation of information on virus host species will allow for adequate future planning of epidemiological measures, suggest the most likely routes of spread of the virus, and assist in the prediction of the introduction of the virus in the western coastal region of the middle Caspian Sea [40].

## Figures and Tables

**Figure 1 viruses-16-00598-f001:**
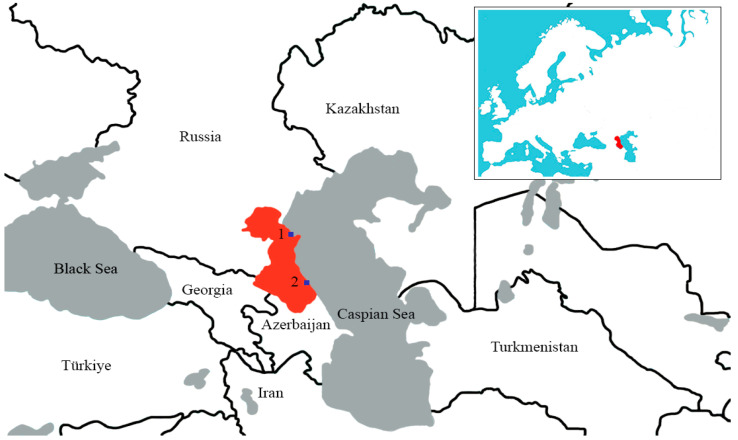
Sampling sites in the Republic of Dagestan (red color): 1—the delta of the Terek River; 2—Lake Adzhi (Papas).

**Figure 2 viruses-16-00598-f002:**
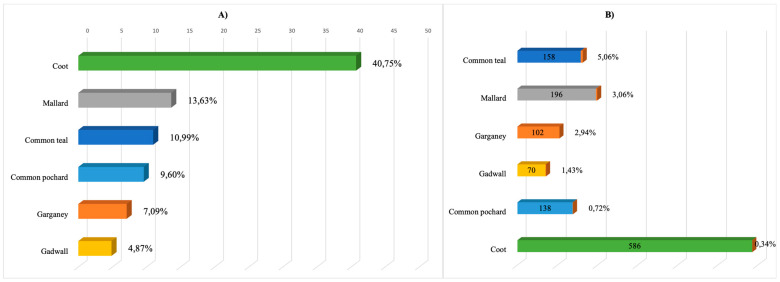
Sample size and prevalence of influenza A virus. (**A**) The most represented bird species in the samples of the 2017–2020 period; (**B**) prevalence of influenza A virus among birds of the 2017–2020 period.

**Figure 3 viruses-16-00598-f003:**
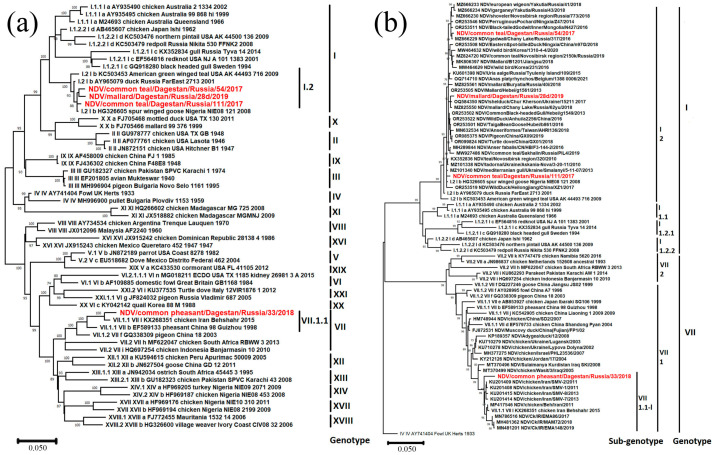
Common phylogenetic tree of the Newcastle disease virus F gene. (**a**) Phylogenetic analysis of NDV Class II isolates, based on the complete F gene CDS (1662 nt) constructed with the maximum likelihood method, based on the Tamura 3-parameter model with a discrete gamma distribution (+G) model in MEGA X. The percentage of trees in which the associated taxa clustered together in the bootstrap test (1000 replicates) is shown next to the branches. Isolates used in this study are shown in red. Roman numerals indicate the corresponding genotype and sub-genotype of each isolate according to the classification proposed by Dimitrov et al. [29]. (**b**) Detailed phylogenetic analysis of NDV genotype I and VII isolates with the identically closest strains based on the complete F gene CDS (1662 nt) constructed with the maximum likelihood method, based on the Tamura 3-parameter model with a discrete gamma distribution (+G) model in MEGA X. The percentage of trees in which the associated taxa clustered together in the bootstrap test (1000 replicates) is shown next to the branches. Isolates used in this study are shown in red. Roman numerals indicate the corresponding genotype and sub-genotype of each isolate according to the classification proposed by Dimitrov et al. [29]. The tree scale bar represents the number of substitutions per sites.

**Figure 4 viruses-16-00598-f004:**
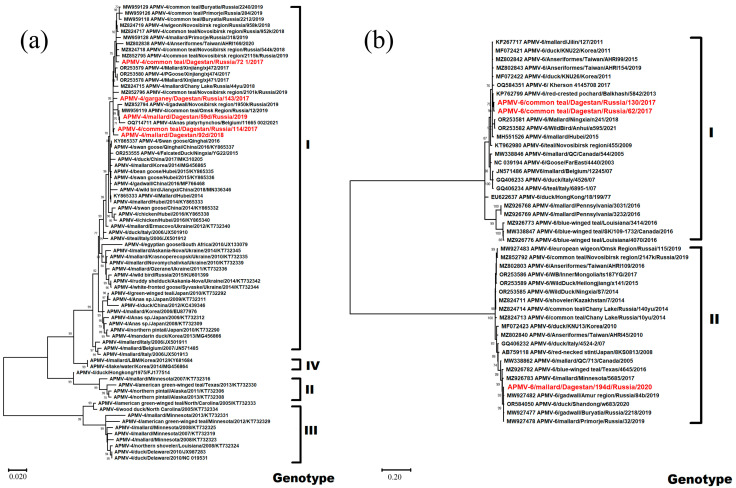
Phylogenetic tree of the Avian paramyxovirus 4,6 F gene. (**a**) Phylogenetic analysis of Avian paramyxovirus 4 isolates based on the complete F gene CDS (1701 nt) constructed with the maximum likelihood method, based on the Tamura 3-parameter model with invariant sites (+I) model in MEGA X. The percentage of trees in which the associated taxa clustered together in the bootstrap test (1000 replicates) is shown next to the branches. Isolates used in this study are shown in red. Roman numerals indicate the corresponding genotype. (**b**) Phylogenetic analysis of Avian paramyxovirus 6 isolates based on the complete F gene CDS (1668 and 1638 nt) constructed with the maximum likelihood method, based on the Tamura 3-parameter model with a discrete gamma distribution (+G) and allowing for invariant sites (+I) in MEGA X. The percentage of trees in which the associated taxa clustered together in the bootstrap test (1000 replicates) is shown next to the branches. Isolates used in this study are shown in red. Roman numerals indicate the corresponding genotype. The tree scale bar represents the number of substitutions per sites.

**Figure 5 viruses-16-00598-f005:**
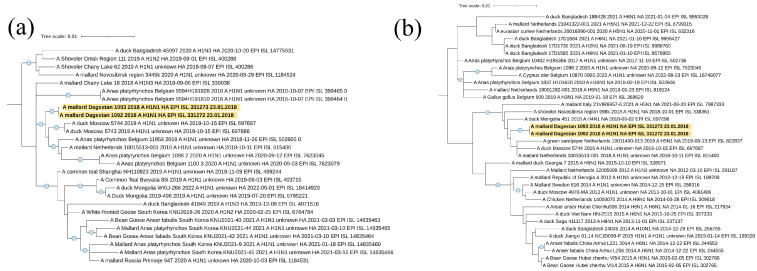
Phylogenetic analysis of H1N1 avian influenza viruses. (**a**) Maximum likelihood phylogenetic tree of the HA (H1) genome segment of avian influenza viruses isolated in the west Caspian region (2017–2020). (**b**) Maximum likelihood phylogenetic tree of the NA (N1) genome segment of avian influenza viruses, isolated in the west Caspian sea region samples (2017–2020). In both figures, the blue circle symbol denotes branches with values SH-aLRT > 80% and UFboot > 95%. The tree scale represents the number of substitutions per site.

**Figure 6 viruses-16-00598-f006:**
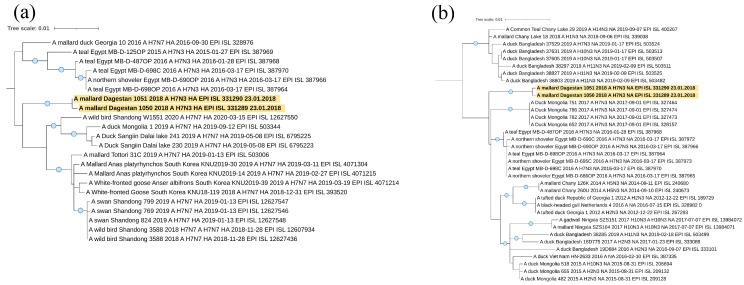
Phylogenetic analysis of H7N3 avian influenza viruses. (**a**) Maximum likelihood phylogenetic tree of the HA (H7) genome segment of avian influenza, isolated in the western Caspian region samples (2017–2020). (**b**) Maximum likelihood phylogenetic tree of the NA (N3) genome segment of avian influenza viruses, isolated in the western Caspian region samples (2017–2020). In both figures, the blue circle symbol denotes branches with values SH-aLRT > 80% and UFboot > 95%. The tree scale represents the number of substitutions per site.

**Figure 7 viruses-16-00598-f007:**
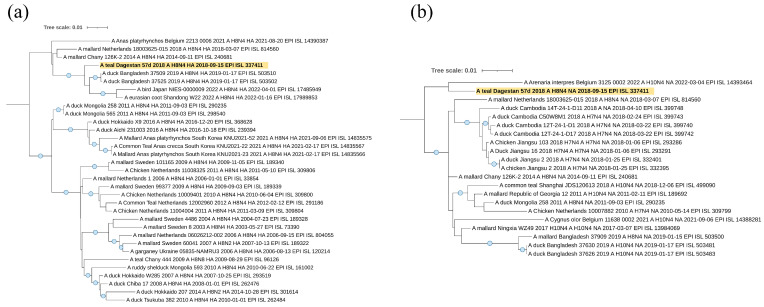
Phylogenetic analysis of H8N4 avian influenza viruses. (**a**) Maximum likelihood phylogenetic tree of the HA (H8) genome segment of avian influenza virus, isolated in western Caspian region samples (2017–2020). (**b**) Maximum likelihood phylogenetic tree of the NA (N4) genome segment of avian influenza virus, isolated in western Caspian region samples (2017–2020). In both figures, the blue circle symbol denotes branches with values SH-aLRT > 80% and UFboot > 95%. The tree scale represents the number of substitutions per site.

**Figure 8 viruses-16-00598-f008:**
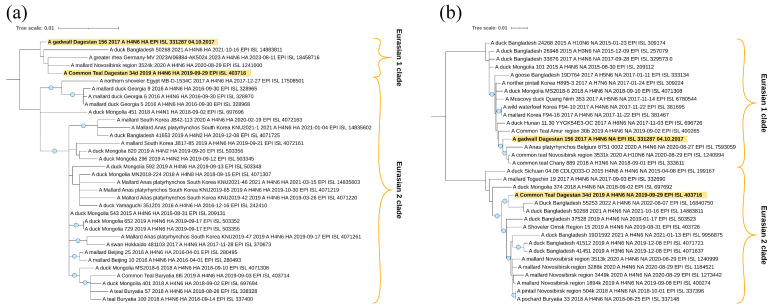
Phylogenetic analysis of H4N6 avian influenza viruses. (**a**) Maximum likelihood phylogenetic tree of the HA (H4) genome segment of avian influenza viruses, isolated in western Caspian region samples (2017–2020). (**b**) Maximum likelihood phylogenetic tree of the NA (N6) genome segment of avian influenza viruses, isolated in western Caspian region samples (2017–2020). In both figures, the blue circle symbol denotes branches with values SH-aLRT > 80% and UFboot > 95%. The tree scale represents the number of substitutions per site.

**Figure 9 viruses-16-00598-f009:**
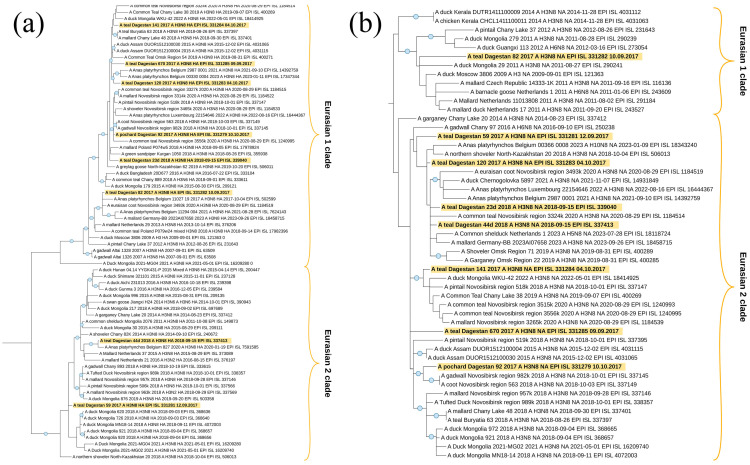
Phylogenetic analysis of H3N8 avian influenza viruses. (**a**) Maximum likelihood phylogenetic tree of the HA (H3) genome segment of avian influenza viruses, isolated in western Caspian region samples (2017–2020). (**b**) Maximum likelihood phylogenetic tree of the NA (N8) genome segment of avian influenza viruses, isolated in western Caspian region samples (2017–2020). In both figures, the blue circle symbol denotes branches with values SH-aLRT > 80% and UFboot > 95%. The tree scale represents the number of substitutions per site.

**Figure 10 viruses-16-00598-f010:**
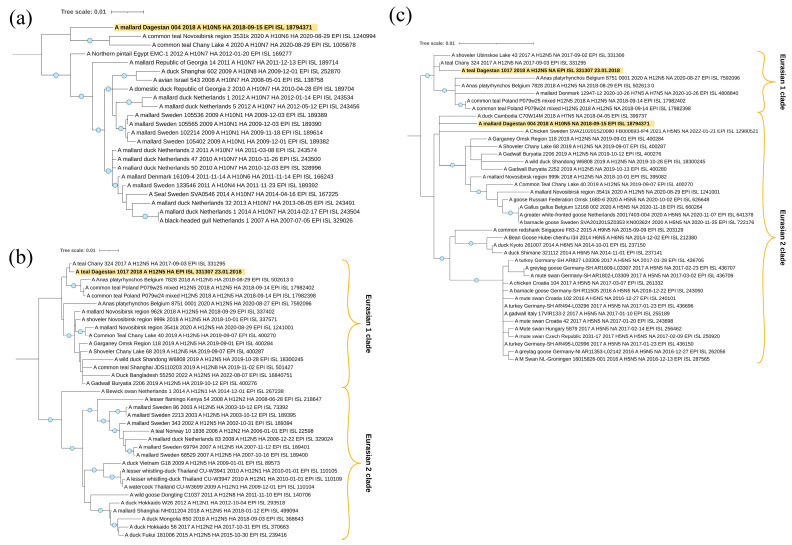
Phylogenetic analysis of H12N5 and H10N5 avian influenza viruses. (**a**) Maximum likelihood phylogenetic tree of the HA (H10) genome segment of avian influenza virus, isolated in western Caspian region samples (2017–2020). (**b**) Maximum likelihood phylogenetic tree of the HA (H12) genome segment of avian influenza virus, isolated in western Caspian region samples (2017–2020). (**c**) Maximum likelihood phylogenetic tree of the NA (N5) genome segment of avian influenza viruses, isolated in western Caspian region samples (2017–2020). In the three figures, the blue circle symbol denotes branches with values SH-aLRT > 80% and UFboot > 95%. The tree scale represents the number of substitutions per site.

**Table 1 viruses-16-00598-t001:** Sample size and results of virus detection in wild waterfowl of the western coast of the middle Caspian Sea.

Order	Species	Numbers of Samples in Periods			Total Number	Number of HA Viruses	Number of AIV	Number of APMV	Virus Isolation Rate (CI 95%)
		1	2	3					
Anseriformes *n =* 797	bean goose (*Anser fabalis*)	0	1	0	1	-	0	0	0
	Greylag goose (*Anser anser*)	3	0	1	4	0	0	0	0
	common pochard (*Aythya ferina*)	51	70	17	138	1	1	0	0.72% (0.02–3.97%)
	red-crested pochard (*Netta rufina*)	0	1	7	8	1	0	1	12.50% (0.32–52.65%)
	mallard (*Anas platyrhynchos*)	90	60	46	196	10	6	4	5.10% (2.47–9.18%)
	greater scaup (*Aythya marila*)	0	2	0	2	0	0	0	0
	common shelduck (*Tadorna tadorna*)	4	2	4	10	0	0	0	0
	European wigeon (*Anas penelope*)	12	6	10	28	0	0	0	0
	gadwall (*Anas strepera*)	33	15	22	70	1	1	0	1.43% (0.04–7.70%)
	tufted duck (*Aythya fuligula*)	15	11	3	29	1	0	1	3.45% (0.09–17.76%)
	common teal (*Anas crecca*)	121	5	32	158	13	8	4	8.23% (4.45–13.66%)
	garganey (*Anas querquedula*)	31	37	34	102	8	3	3	7.84% (3.45–14.87%)
	northern pintail (*Anas acuta*)	11	6	8	25	1	0	1	4% (0.10–20.35%)
	shoveler (*Anas clypeata*)	7	8	11	26	0	0	0	0
Gruiformes *n* = 588	coot (*Fulica atra*)	73	254	259	586	6	2	0	1.02% (0.38–2.23%)
	common moorhen (*Gallinula chloropus*)	0	2	v	2	0	0	0	0
Podicipediformes *n* = 2	great crested grebe (*Podiceps cristatus*)	0	0	1	1	0	0	0	0
	little grebe (*Tachybaptus ruficollis*)	0	0	1	1	0	0	0	0
Galliformes *n* = 1	common pheasant (*Phasianus colchicus*)	1	0	0	1	1	0	1	100% *
Pelecaniformes *n* = 8	great cormorant (*Phalacrocorax carbo*)	2	4	0	6	0	0	0	0
	grey heron (*Ardea cinerea*)	1	0	1	2	0	0	0	0
Charadriiformes *n* = 38	black-headed gull (*Larus ridibundus*)	0	0	25	25	0	0	0	0
	common snipe (*Gallinago gallinago*)	9	0	1	10	0	0	0	0
	wood sandpiper (*Tringa glareola*)	1	0	0	1	0	0	0	0
	black-tailed godwit (*Limosa limosa*)	2	0	0	2	0	0	0	0
Ciconiiformes *n* = 4	great bittern (*Botaurus stellaris*)	0	0	4	4	0	0	0	0
7 orders	26 species	467	484	487	1438	43	21	15	-

CIs were determined using the Clopper-Pearson method; *—not reliable.

**Table 2 viruses-16-00598-t002:** Isolates of avian paramyxoviruses obtained in this study.

Isolate	Host	Class	Genotype/ Sub-Genotype	Cleavage Site of Fusion Protein	Year of Isolation	GenBank Accession No.
NDV/common teal/Dagestan/Russia/111/2017	common teal	II	I.2	^112^ GKQGR↓L ^117^	2017	MZ666236
NDV/common teal/Dagestan/Russia/54/2017	common teal	II	I.2	^112^ GKQGR↓L ^117^	2017	PP537563
NDV/common pheasant/Dagestan/Russia/33/2018	common pheasant	II	VII.1.1	^112^ RRQKR↓F ^117^	2018	PP537562
NDV/mallard/Dagestan/Russia/28d/2019	mallard	II	I.2	^112^ GKQGR↓L ^117^	2019	MW927498
APMV-4/common teal/Dagestan/Russia/72_1/2017	common teal	-	I	^116^ DIQPR↓F ^121^	2017	PP537558
APMV-4/common teal/Dagestan/Russia/114/2017	common teal	-	I	^116^ DIQPR↓F ^121^	2017	PP537557
APMV-4/garganey/Dagestan/Russia/143/2017	garganey	-	I	^116^ DIQPR↓F ^121^	2017	PP537556
APMV-4/mallard/Dagestan/92d/2018	mallard	-	I	^116^ DVQPR↓F ^121^	2018	MW880773
APMV-4/mallard/Dagestan/59d/Russia/2019	mallard	-	I	^116^ DIQPR↓F ^121^	2019	MZ852793
APMV-6/common teal/Dagestan/Russia/62/2017	common teal	-	I	^114^ VPEPR↓L ^119^	2017	PP537560
APMV-6/common teal/Dagestan/Russia/130/2017	common teal	-	I	^114^ VPEPR↓L ^119^	2017	PP537561
APMV-6/mallard/Dagestan/194d/Russia/2020	mallard	-	II	^104^ IREPR↓L ^109^	2020	PP537564

## Data Availability

All genome sequences of AIVs from this study are available in GISAID database (accession numbers: EPI_ISL_331289, EPI_ISL_331290, EPI_ISL_331307, EPI_ISL_337411, EPI_ISL_337413, EPI_ISL_3390406, EPI_ISL_403716, EPI_ISL_18794371, EPI_ISL_3312876, EPI_ISL_331279, EPI_ISL_331283, EPI_ISL_331284, EPI_ISL_331281, EPI_ISL_331285, EPI_ISL_331282, EPI_ISL_331272, EPI_ISL_331273). All genome sequences of APMVs from this study are available in GeneBank (accession numbers: PP537563, PP537562, MW927498, PP537558, PP537557, PP537556, MW880773, MZ852793, PP537560, PP537561, PP537564, MZ666236).

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
