# Peer review of "Avian Influenza Virus and Avian Paramyxoviruses in Wild Waterfowl of the Western Coast of the Caspian Sea (2017–2020)"

_viruses, 2024, doi:10.3390/v16040598_

Round 1

Reviewer 1 Report

Comments and Suggestions for Authors

The authors conducted wild bird surveillance for avian influenza and “avian paramyxovirus” on the western coastline of the Caspian Sea, which falls within an important flyway for birds moving to and from other continents. The full genome sequences of 36 avian influenza virus isolates and 15 “avian paramyxoviruses” are a valuable contribution to the international databases. The study analyses the distribution of avian influenza subtypes detected in different aquatic bird species to identify key hosts, contributing valuable ecological data too. The clinical samples collected during surveillance were blindly inoculated into eggs for virus isolation, and hemagglutinating agents were analyzed further by virus-specific RT-PCR screening and genome sequencing. The subsequent phylogenetic analysis focused on avian influenza, and the analysis of “avian paramyxoviruses” was not presented. For avian influenza viruses, only the phylogenetic analysis of the internal proteins was presented, but not the HA and NA antigens, which would be of greater scientific value and interest. Somewhat bizarrely, intravenous pathogenicity tests were performed on five H1N1 and H3N8 viruses (low pathogenicity), with predictable results.  It is odd that no H5Nx HPAI viruses were identified given the strategic geographic location of the sampling sites in a region that is constantly affected by outbreaks. The manuscript does require further English language editing. Specific recommendations are made below.

Specific comments

Lines 26, 79 and elsewhere: Please change “highly pathogenic avian influenza” to “high pathogenicity avian influenza”.

Lines 84-87 and elsewhere: Please check ICTV sources, “avian paramyxovirus” was renamed as avian avulavirus several years ago, and make changes to the nomenclature throughout.

Line 101: Please clarify- the swabs were collected from the carcasses of dead birds? Were these freshly shot?

Line 102: What did the viral transport medium consist of? Please specify.

Lines 126-129: This paragraph does not fit here, rather move it to after genome sequencing, presumably the subtypes were determined by BLAST analysis of assembled HA and NA sequences? Please clarify. Also, please specify the accession numbers of the sequences that were used for assembly.

Line 152: change OIE to WOAH

Line 156: “no unexpected death or clinical signs were observed” move to Results section

Lines 162-164: How did you assemble the avian influenza and avian paramyxovirus genome sequences? What software was used? Please provide this important missing information.

Table 1: heading- change “number total” to “total number”.

Line 245: change “isolates were isolated” to “isolates were obtained”

Lines 252-253: You used genome sequencing (which has the wonderful benefit of being non-specific).  Please explain why seven of the non-hemagglutinating viruses could not be identified? Did you try a de novo assembly of the sequence reads and BLAST? The details for how you actually assembled your AIV and APMV genomes are missing, and must be added. With deep sequencing of isolated viruses, there is no good reason why they could not be identified, and the same applies for the “HxNx” AIVs.  

Line 259: HxNx- I find it hard to believe that the subtypes could not be identified from viruses cultured in alantoic fluid (containing large amounts of vRNA) subjected to deep sequencing. Please explain. Did the sequencing not work for these cases? No information on the quality of the sequencing results (depth of coverage etc.) is provided.

Line 259: What was the HA cleavage site motif in the H7N3 viruses? Were they confirmed as low pathogenicity based on the sequence? Please specify.  

Lines 289-290: “In the context of the segmentation of the genome of the virus central to this study, phylogenetic patterns were analysed internal segments”. Considering the data presented, this statement should be contained in the purpose of the study on page 1. However, you cannot avoid presenting the HA and NA phylogenetic sequence data, without it this study is incomplete. Your phylogenetic findings with the internal genes are fairly standard and predictable, and doesn’t contribute novel data. For this reason, I suggest you add all the trees for the HA genes into the body of this manuscript and briefly discuss them because the surface glycoproteins the main determinants of pathogenicity and antigenically relevant for vaccines and as diagnostic reagents. This would be of greater interest to international scientists than the analysis of the internal genes. All the current phylogenetic trees should be incorporated as supplemental data, and summarized into a single paragraph (briefly, all belong to Eurasian lineage, a few associated with HPAI strains, both NS1 alleles detected). Furthermore, I recommend check the amino acid sequences of all genes for known markers of mammalian adaptation to further strengthen the manuscript.

Lines 414-419: please add virus subtypes to the names. It isn’t clear why you performed IVPIs on viruses that were clearly LPAI. Yes, the internal genes may contribute to pathogenicity, but there was no rational reason given (e.g., the HA cleavage motifs were highly unusual; suspect molecular markers of pathogenicity/virulence/host adaptation were present; most of the internal gene constellation was associated with and HPAI virus that caused massive outbreaks… etc.) to sacrifice chickens in this case.

Finally, either you need to add all the phylogenetic analysis for the APMVs isolated and sequenced, or state somewhere that the data will be presented elsewhere (i.e., not part of the present study) - and update the heading accordingly.  

Comments on the Quality of English Language

Please revise spelling and grammar throughout, I have not flagged all of the changes required (e.g., correct "Analysis" in the subheadings).

Reviewer 2 Report

Comments and Suggestions for Authors

This study described the detection and phylogenetic analyses of avian influenza viruses, as well as, avian paramyxovirus, from wild bird of western coast of Caspian sea during autumn –winter periods (years 2017 to 2020). This study is interesting because this geographical region is an important wetland where many bird species congregate and is an important route for certain migratory birds. However, in my opinion, the introduction needs to be reoriented and some elements of the materials and methods need to be clarified.

Title :

The title used here indicates only a detection for avian influenza virus whereas a detection of avian paramyxovirus was also performed.

Introduction :

I think that the introduction is a little too focused on crossing the species barrier, detection in mammals and zoonotic risk, whereas this introduction almost overlooks the description of avian paramyxoviruses, which are also being investigated in this study.

Materials and methods :

-          The tested birds were hunting birds? birds caught alive? Or collected dead birds?

-          Line 110 : add “…to isolate AIVs and avian paramyxovirus”

-          Line 117 : please replace for “AIV M gene PCR testing” by “AIV and APMV detection by RT-PCR

-          Line 121 : replace “isolated” by “extracted”

-          Line 124 : Is the kit A/B FL kit is no more adapted to detect human influenza than to the avian influenza virus

-          How the HA and NA were amplified and sequenced. If it s simultaneous with the complete genome. Please, add this part in the “sequencing and phylogenetic analysis” paragraph

-          For the sequencing, authors used Nextera XT DNA, but how the DNA sequences were obtained? Which RT was used?

-          In the paragraph 2.4.2, how the type of the APMV was determined? In Addition, in the supplementary data, sometimes the type of APMV was indicated sometimes not, why?

Results:

-          Line 196/199: please, replace “seasons” by “periods”

-          Line 245 : “36 isolates” seems to be correspond to a part of the AIV isolates but it’s was also indicate that only 21 isolates correspond to AIV. SO which was the good number?

-          Line 291/306/325/340/367/388: please change “Analisys” by “Analysis”

-          For phylogenetic trees: please indicate the units of the scale

-          Line 295: may be the term: “closely related” is more appropriate than “consisting”

-          Please standardizes the term “clade” or “group” between the text and the phylogenetic trees

-          Line 402 : The two groups detected in the NS B allele were not clearly exposed in the figure 8

-          No result about APMV typing were showed in this article. How authors known when it is APMV4 or not (indicated in Table S1)?

Discussion:

-          The authors point out that the Caspian Sea region is probably a genetic crossroads between Europe, Asia and, to a lesser extent, Africa. This is probably linked to the migratory routes of wild birds that pass through this region.

But in my opinion, there are other elements missing from the discussion, such as :

- comparison of the prevalence with other world regions with a high density of wild birds

- Explain why there are identified non-AIV and non-APMV hemagglutinating viruses.

- Why are some AIV and APMV not typed? Methodology?

The use of sequencing methods directly from field samples (without viral isolation) as described by Zhou (Single -reaction genomic amplification accelerates sequencing and vaccine production for classical and swine origin human influenza) and commonly used might have been more efficient and less cumbersome in terms of egg isolation.

Round 2

Reviewer 1 Report

Comments and Suggestions for Authors

Most of the suggested changes were made, and where they were not, justifications were provided. Please attend to language editing. 

Comments on the Quality of English Language

New sections added to the manuscript require substantial editing.

Line 128: “isolation of AIVs”, add a comma after “APMVs”

Line 129: delete “transferred to”, replace “centrifugation” with “centrifuged”

Line 130: put “penicillin and gentamycin” in brackets... “and transferred”

Line 131: replace “Further” with “next”. Rephrase the rest of this sentence: Next, 100ul of each sample…

Line 134: replace text with “2 ml of each alantoic fluid was extracted”

Line 136: write out abbreviations the first time they are used: RDRP

Line 142: replace “conservative” with “conserved”

Line 144: replace “adapted” with “developed”

Line 146: RT-PCR (it’s an RNA virus).

Line 160: For the intravenous…, add a comma after “test”

Line 161: add a comma after [17]

Line 163: use superscript and subscript notations where applicable

Line 167: laboratory in lower case

Line 170: add a comma after 2018

Line 177: de novo should be in italics

Line 192: “a number of”- specify exactly how many

Line 206: move the bracket behind “method” to behind “likelihood”.

Line 220 and elsewhere (change this everywhere please): “complete CDS gene F” should be complete F gene CDS, and define CDS the first time it is used.

Figures 3(a) and (b): please adjust the following fonts (make them smaller and uniform in size) to improve the aesthetics: the letters a) and b), and labels I and VII.

Line 365-366 and elsewhere: move “isolate” to before the strain name (NDV/common teal….)

Line 368 and elsewhere: replace “turned out to be” with “were”

Line 381: 4 and 6

Line 388: Delete “And”, start the sentence with “Only”

Line 406: Delete “As a result of the study”

Line 412: correct spelling of apathogenic

Line 415: The F protein cleavage site 104-IREP…. (change the order please)

Line 433 and elsewhere: amino acid mutations…

Line 435: why is PB2 “most important”? Modify or add a reference to support the statement.

Line 451 and elsewhere: change “According to the neuraminidase N1 segment” to “The N1 neuraminidase gene sequence belonged…”. The use of “segment” is irrelevant, unless you specify segment 5 which encodes for the neuraminidase gene.

Line 468: add a space after “low”

Line 471: been described.

Line 499: change to “suggests a wider connection with”

Line 533: replace “has” with “is of”

Lines 551-558: this should be paragraph text, not figure legend.

Reviewer 2 Report

Comments and Suggestions for Authors

The recommendations from reviwers have been taken into account
